# Relevance and Feasibility of Group Traumatic Episode Protocol Delivered to Migrants: A Pilot Field Study

**DOI:** 10.3390/ijerph20075419

**Published:** 2023-04-06

**Authors:** Philippe Vignaud, Nicolas Chauliac, Emmanuel Contamin, Sébastien Richer, Cécile Vuillermoz, Jérôme Brunelin, Nathalie Prieto

**Affiliations:** 1Hospices Civils de Lyon, Hôpital Edouard Herriot, Centre Régional du Psychotraumatisme Auvergne Rhône Alpes, F-69437 Lyon, France; 2Hospices Civils de Lyon, Hôpital Edouard Herriot, Cellule D’urgences Medico-Psychologiques, F-69437 Lyon, France; 3INSERM U1028, CNRS UMR5292, PSYR2 Team, Centre de Recherche en Neurosciences de Lyon, Université Claude Bernard Lyon 1, F-69500 Bron, France; 4Research on Healthcare Performance (RESHAPE), INSERM U1290, Université Claude Bernard Lyon 1, F-69008 Lyon, France; 5Independent Researcher, 4 Rue du Viel Renversé, F-69005 Lyon, France; 6Independent Researcher, 19 Rue de la République, F-69600 Oullins, France; 7Institut Pierre Louis d’Epidémiologie et de Santé Publique (IPLESP), INSERM, Department of Social Epidemiology, Sorbonne Université, F-75012 Paris, France; 8Centre Hospitalier Le Vinatier, 95 Boulevard Pinel, F-69500 Bron, France

**Keywords:** post-traumatic stress disorder, eye movement desensitization reprocessing, group traumatic episode protocol, group, migrant, care access

## Abstract

**Highlights:**

**What are the main findings?**
Group Traumatic Episode Protocol (G-TEP) may be efficient to treat PTSD symptoms in migrants.G-TEP displayed a tendency toward a decrease in symptoms of depression in migrants.

**What is the implication of the main finding?**
G-TEP is suitable to improve the access to psychiatric care for migrants.

**Abstract:**

Introduction: Post-Traumatic Stress Disorder (PTSD) and Major Depressive Disorder (MDD) are commonly observed in migrants. Although Eye Movement Desensitization and Reprocessing (EMDR) can be helpful to treat these diseases, it remains difficult to propose EMDR as an individual intervention in help-seeking migrants. Group EMDR, like Group Traumatic Episode Protocol (G-TEP), which was built around the 8 phases of the original EMDR protocol, could offer an effective treatment to a large number of people. It may also be more resource-efficient to provide psychiatric care to migrants. Methods: In this open-label trial, the feasibility and the effectiveness of a 6-session G-TEP intervention was investigated in a group of 10 migrants. Results: The intervention was well tolerated by participants. The final attrition rate was 10%. After the intervention, there was a 28.2% significant decrease in PTSD and complex PTSD symptoms, as measured by the International Trauma Questionnaires (total_ITQ) scores (*p* = 0.013) and a trend towards a significant decrease in MDD symptoms, as measured with the Patient Health Questionnaire (PHQ-9) (*p* = 0.057). Conclusions: G-TEP may be effective in decreasing PTSD symptoms in migrants. The accessibility, low-cost, and very structured features of G-TEP may make its implementation sustainable in the field of psychiatric care for migrants.

## 1. Introduction

Migration is a complex and stressful experience. Migrants often experience potential traumatic events (PTE) either in their home country (pre-migration), during migration, and/or after their arrival in the host country (post-migration). Migration and resettlement have been associated with various mental-health problems and psychiatric conditions, the prevalence of which varies across studies. Among the most common psychiatric disorders associated with migration, Post-Traumatic Stress Disorder (PTSD) (prevalence varying between 9 and 36% across studies) and Major Depressive Disorder (MDD) (prevalence varying between 5 and 44% across studies) raise particular concern, as they are associated with long-lasting functional impairment [1,2,3] Although it is difficult to diagnose and provide appropriate health care due to language and cultural differences in these populations, there is a need to develop effective tailored treatments for these disorders to overcome the poor prognosis.

Eye Movement Desensitization and Reprocessing (EMDR) is a psychotherapeutic approach, which was designed on the basis of the so-called adaptive information processing (AIP) model [4]. By involving exposure to PTE memories or otherwise adverse life experiences, EMDR may restore the AIP, so that the distressing memories can be processed and functionally stored. Notably, the reprocessing phase of EMDR includes the application of bilateral stimulations (BLS), most typically consisting in sets of rapid eye movements, while the patient is asked to focus on a distressing memory. In several systematic reviews and meta-analyses, EMDR was found to be effective in treating certain psychiatric disorders, especially PTSD [5,6,7] depressive symptoms [8], and anxiety disorders [9]. Several worldwide guidelines recommend EMDR as an effective treatment for PTSD [10,11,12].

EMDR was designed and used as an individual intervention in all the above-mentioned studies. Since clinicians require complementary training to apply this treatment, individual EMDR may not be the most time- and cost-efficient psychotherapy when a high number of patients need treatment in a short time frame. This is a critical issue when considering access to mental health care for migrants, especially in Europe where a large proportion of migrants who have arrived over the last years have been exposed to PTE. The high volume of mental health care needed and the large number of potential patients among migrants are challenging the access to mental health care systems in host countries [13]. 

Thus, group EMDR, which could offer an effective treatment to a large number of people, may be more resource-efficient. Up to now, several EMDR group protocols have been developed, including Group Traumatic Episoderotocol (G-TEP) [14,15]. The tool is built around the eight phases of the original EMDR protocol, and a special worksheet enables the practitioner to make every participant follow these different phases at a group level. During a preliminary step, the participants learn and practice exercises of emotional stabilization. Mainly, these exercises provide the participants with some tools to better cope with stress or negative emotions, such as abdominal breathing and mindfulness-based exercises. They are then taught to identify a so-called Point of Disturbance (PoD), as if they would search for therapeutic targets in the usual individual EMDR protocol. The Subjective Units of Disturbance (SUD) (an analogic scale (min 0–max 10) assessing the level of the induced psychological distress) level of the target PoD is assessed, and the PoD is processed using BLS including bilateral eye movements. Each participant taps with his/her own hand from one spot of the worksheet, the present safety, to another spot on the worksheet, the PoD. While tapping, they follow their hand with the eyes, thus self-generating BLS. Every three sets of BLS, the participants focus on the PoD and assess the SUD again. After nine sets, they look for a new PoD, and they process it in the same way. Three PoDs are processed in each G-TEP session. Then, the participants think of a Positive Cognition (PC). Finally, the session ends with a stabilization exercise. A safety assessment is included at the very beginning of the protocol to identify the participants who are not ready for trauma processing. Indeed, the trauma processing can induce some emotional distress, and the participants should be stable enough to bear the emotional challenge related to the desensitization process. Extreme emotional reactions should be avoided to preserve the therapeutic process at the group level. Compared to individual EMDR, G-TEP has several advantages. First, whether or not the participants know each other before attending the G-TEP protocol, a group cohesion often appears and encourages each participant to maintain session attendance. This is facilitated by the fact that communication between participants during the session focuses only on the positive feedback of the stabilization exercises and excludes any sharing concerning traumatic memories. Secondly, as the participants stay quiet while processing the PoD, high language skills are not required. This is a critical aspect for participants who do not speak the same language within the group or have difficulties with the language of the host country.

Therefore, group EMDR has been gaining attention in the scientific literature in the past few years [16], highlighting promising results to alleviate MDD and PTSD symptoms. However, only a few studies evaluating the effect of group EMDR on migrants have been conducted. Most of them included children, and an adapted version of group EMDR for this pediatric population was delivered [17,18]. To the best of our knowledge, the effect of G-TEP in migrants was evaluated in only two previous studies [19,20]. Therefore, the present pilot study aimed to further explore the feasibility of delivering G-TEP in migrants and to assess its therapeutic potential concerning MDD and PTSD symptoms in that specific population.

## 2. Methods

This open-label pilot study investigated the clinical effects of G-TEP in help-seeking migrants. First, all participants had an interview with a psychiatrist (P.V.) to control for group eligibility. The inclusion criteria were: PTSD and/or MDD (according to the DSM-5 criteria), the possibility of attending all sessions of the G-TEP protocol, and sufficient emotional stability to process traumatic memories. The exclusion criteria were: comorbid diagnosis of schizophrenia, schizoaffective disorder, bipolar disorder, eating or obsessive-compulsive disorder, substance use disorder, borderline personality disorder, or any neurological disorder. Participants were asked to complete two self-assessment questionnaires: the 12-item International Trauma Questionnaire (ITQ) and the 9-item version of the Patient Health Questionnaire (PHQ-9). During the last session of the G-TEP intervention, ITQ and PHQ-9 were assessed again to evaluate changes.

### 2.1. Participants

The participants were migrants identified as suffering from symptoms of PTSD and/or MDD and referred to our department. We have established a partnership with a charity that helps migrants solve their administrative issues and offers them social support. The sample included 10 migrants coming from various continents (Europe, Asia, and Africa): 7 women and 3 men (mean age 31.3 years old; standard deviation SD = 11.9; range 17–53) (Table 1). The G-TEP intervention took place between May and June 2021.

### 2.2. Clinical Assessment

The ITQ is a self-reported short and simply-worded measure, focusing on the key symptoms of PTSD and complex-PTSD (CPTSD). The total_ITQ score is composed of a PTSD score and a Disturbances in Self-Organization (DSO) score. The PTSD score is a Likert scale composed of 6 items (P1–P6); the P1 and P2 items represent the re-experiencing-in-the-here-and-now score, P3 and P4 the avoidance score, and P5 and P6 the sense of a current threat. The DSO score is a Likert scale also composed of 6 items (C1–C6); C1 and C2 represent affective dysregulation, C3 and C4 negative self-perception, and C5 and C6 disturbances in relationships. The functional impact of these symptoms was also measured using the sum of the P7–P9 and C7–C9 items of the ITQ (functional_ITQ).

The Patient Health Questionnaire (PHQ-9) is composed of 9 items on a Likert scale, assessing symptoms of MDD. Each participant was accompanied by a volunteer of the charity. For the non- or partial-French-speaking migrants, the volunteer translated, or at least helped the participant understand the inclusion interview.

### 2.3. G-TEP Intervention

The G-TEP intervention consisted of 6 weekly 2-hour sessions. The first 2 sessions were used to train participants on stabilization exercises and for the detailed presentation of the G-TEP worksheet. The following 3 sessions were dedicated to the desensitization process. The last session was dedicated to the self-assessment questionnaires, to a general feedback about the intervention and to a reminder of the stabilization exercises. The group was moderated by two psychiatrists (E.C. and P.V.) and a psychologist (S.R.). While one moderator was effectively leading the group intervention, the two other moderators checked that every participant understood the instructions and remained active. They could also more intensively support a participant in case of abreaction. Non-French-speaking participants were given a worksheet translated in their native language (1 in English, 1 in Arabic).

### 2.4. Statistical Analysis

The primary outcome measures were the ITQ sub-scores (total_ITQ scores and functional_ITQ scores) and the PHQ-9 scores completed at baseline and after the completion of the 6 sessions. The Wilcoxon signed rank test was used to compare the pre- and post-intervention scores. Statistical analyses were performed using JASP (Version 0.16.3) [Computer software] (JASP Team (2022), Amsterdam, The Netherlands). The PTSD and DSO scores measured using the ITQ were also compared pre- and post-intervention.

## 3. Results

All participants except three attended all six sessions of the G-TEP intervention. The two 17-year-old women from Cameroon (Participants 6 and 7) missed the 4th session. Another participant, a 22-year-old man from Guinea (Participant 10), missed the last session and had to be excluded from the analysis because of missing post-intervention data. Participants tolerated the intervention well, with the exception of one patient (Participant 7) who reported several abreactions during the procedure. Detailed results are given in Table 2.

At the end of the G-TEP intervention, there was a 28.2% significant decrease in PTSD and CPTSD symptoms, as measured by the total_ITQ scores (W = 44.0; z = 2.547; *p* = 0.013; effect size measured by rank Biserial correlation = 0.956, 95%CI [0.820; 0.990]). Regarding the ITQ sub-scores, there was a 41% significant decrease in DSO scores (W = 45.0; *p* = 0.009) and a 15% non-significant decrease in PTSD scores (W = 28.5; *p* = 0.16). There was also a 42.6% significant decrease concerning the functional impact of PTSD symptoms, as measured by the sum of the ITQ P7–P9 and C7–C9 items (W = 33.0; z = 2.100; *p* = 0.042; effect size = 0.833, 95%CI [0.398; 0.962]). There was a non-significant 28.4% decrease in depressive symptoms (assessed by the PHQ-9) after the G-TEP intervention (W = 39.0; z = 1.955; *p* = 0.057; effect size = 0.733, 95%CI [0.198; 0.932]).

## 4. Discussion

In this open-label pilot study, G-TEP was found to significantly decrease symptoms of PTSD, especially those relating to disturbances in self-organization, in a group of migrant participants who completed the intervention. This improvement was accompanied by a significant decrease in the functional impact of PTSD symptoms. Symptoms of MDD also decreased, although non-significantly. Notably, this decrease almost reached statistical significance (*p* = 0.057), so that this result appears promising. These results are consistent with previously published studies reporting a beneficial effect of G-TEP intervention on both PTSD and depressive symptoms in migrants. Lehnung et al. [19] reported a 50% decrease in PTSD symptoms assessed with the Impact of Event Scale-Revised (IES-R) in the active group (baseline: mean = 41.8, SD = 15.6; post-intervention: mean = 21.6, SD = 9.9). Yurtsever et al. [20] also found a 20% decrease in the PTSD symptoms assessed with the IES-R in the group who received the GTEP intervention (difference = 14.22, SE = 4.81; *p* < 0.05). Altogether, these results suggest that G-TEP intervention may be effective in mixed groups of migrants regardless of age, culture, home country, and native language. As G-TEP is a structured and time-limited intervention, it may be a powerful therapeutic tool for the psychiatric care of migrants and refugees provided by host countries.

The proposed intervention consisted of 6 weekly 2-hour sessions. This framework is consistent with the one used in other studies investigating the clinical interest of G-TEP EMDR (for a review, see [20]) in terms of length, number, and duration of sessions. Although six sessions may be considered demanding for the participants, two sessions were used to train participants on stabilization exercises and for the detailed presentation of the G-TEP worksheet, which should be viewed as important preliminary steps before the desensitization process. As many migrants have a high exposure to PTE, one single session of desensitization is probably not sufficient.

Two specificities of the intervention proposed herein should be discussed. First, the sessions were conducted with a high number of practitioners, two psychiatrists and a psychologist. It may have been possible to deliver this intervention with only two practitioners. Secondly, each participant was accompanied by a volunteer who was always the same throughout the intervention. This was done to provide social support during the difficult desensitization process and to encourage session attendance. Since only three participants missed only one session each, these objectives can be considered as having been achieved, and the final attrition rate was 10%. The support provided by the volunteers was intensive, and this may have influenced the results concerning the GTEP intervention. Nevertheless, because of social, administrative, and cultural issues they face, migrants have more difficulties in accessing health care, and stronger support is necessary.

There are some limitations to the present study. The analyzed sample was small, which drastically decreased the statistical power of the study. Because of the limited number of participants, the diagnostic features of the ITQ could not be used. Moreover, this study did not include a control group, and there was no follow-up assessment after the intervention. Thus, the expectancy bias and the natural evolution of the symptoms were not controlled. There was a large heterogeneity concerning the clinical responses to G-TEP, with clear beneficial effects in some participants (e.g., participants 5 and 9) but no benefits at all for other participants (e.g., participant 3). The size of the sample did not allow us to investigate potentially predictive markers of response.

## 5. Conclusions

G-TEP may thus be effective in decreasing PTSD symptoms in migrants. Although this was not statistically significant in this study, a possible effect on depressive symptoms deserves to be further explored in further investigations. The accessibility, low-cost, and very structured features of G-TEP may be core factors for its implementation in future clinical routine, especially in the field of psychiatric care provided to migrants, which is still an unsolved issue.

## Figures and Tables

**Table 1 ijerph-20-05419-t001:** Demographic characteristics of the included participants.

Participants	Sex (M/F)	Age (Years)	Home Country	French Level —0: no Understanding and no Expression —1: Understood and Spoke French with Help —2: Understood and Spoke French Fluently
Participant 1	F	34	Armenia	1
Participant 2	F	25	Nigeria	0
Participant 3	F	53	Irak	1
Participant 4	M	31	Guinea	2
Participant 5	M	29	Togo	2
Participant 6	F	17	Cameroon	2
Participant 7	F	17	Cameroon	2
Participant 8	F	45	Syria	2
Participant 9	F	40	Albania	1
Participant 10	M	22	Guinea	2

**Table 2 ijerph-20-05419-t002:** Changes in symptoms of post-traumatic stress disorder (PTSD) and depression in participants receiving the 6-session Group Traumatic Event Protocol (G-TEP). *p:* Wilcoxon signed rank test; SD: standard deviation.

	Total_ITQ Baseline	Total_ITQ Post G-TEP	Functional_ITQ Baseline	Functional_ITQ Post G-TEP	PHQ-9 Baseline	PHQ-9 Post G-TEP
Participant 1	29	22	12	12	15	16
Participant 2	38	28	10	11	21	16
Participant 3	14	17	11	12	5	6
Participant 4	35	27	8	4	15	9
Participant 5	37	18	15	3	24	11
Participant 6	30	26	24	3	21	17
Participant 7	42	32	21	16	22	17
Participant 8	12	8	4	2	7	9
Participant 9	29	13	10	3	18	5
Mean	29.56	21.22	12.78	7.33	16.44	11.78
SD	10.38	7.82	6.30	5.34	6.67	4.82
*p*		0.013		0.042		0.057

## Data Availability

For privacy reasons, the data that support the findings of this study cannot be made publicly available. However, they remain available from the corresponding author, PV, upon reasonable request.

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
