# Peer review of "Relevance and Feasibility of Group Traumatic Episode Protocol Delivered to Migrants: A Pilot Field Study"

_ijerph, 2023, doi:10.3390/ijerph20075419_

Round 1
Reviewer 1 Report
Dear authors:
The paper seems very useful and interesting. I only have some minor issues:
In the introduction section:
1.- The definition of SOD (Subjective Units of disturbance) should be more explained
2.- The definition, or a more detailed explanation of "stabilization exercise/emotional stabilization" should be added.
3.- What the authors refer with: "safety assessment"?
4.- Line 11, a space is missing between "P1" and "and"
Methods section:
5.- The Wilcoxon test is not a "T Wilcoxon test" please delete the "T", this test yield a Z score, not a T or W score, so the name is only: "Wilcoxon signed rank test"
6.- Likewise, when presenting the results, Lines: 180-183, please check if the W is correct, or Z is the correct symbol of the Wilcoxon test.
7.- The statistical difference in the PHQ-9 instrument are borderline, this should be emphasized in the results and discussion section, because the authors only mention it as not significant
Discussion section:
8.- I suggest to be more specific when reporting the similitudes with previous reports: references : Lehnung et al., 2017; Yurtsever et al., 2018, this in terms of percentage of improvement in the instruments measured and/or effect sizes.
9.- When the authors report the limitations of the study, in the case of the lack of a control group, it should be mentioned the bias that this implies, i.e.: that the expectancy bias and the natural evolution of the phenomena were not controlled.
Reviewer 2 Report
Overall, I thought this manuscript was well-written and a wonderful addition to the literature. Unfortunately, there are many migrants in the world today, most of whom have experienced trauma. Testing out new ways to treat the consequences of trauma is critical.
This is meant to be a feasibility study - I urge the authors to think a bit more about whether the resources needed to implement this intervention would be feasible in most migrant settings. It seemed pretty resource-intense, for good reason; this is acknowledged in the Discussion, but I would think about more specifically saying that this might deter scalability, and consider calling for further research to explore the threshold of resources needed to be successful.
As a pilot, I think the authors did a nice job showing feasibility, but there are many additional questions that deserve exploration in future research. I believe the authors understand this, but I would point some of them out more explicitly:
· * Do the authors have any thoughts on whether fewer desensitization process sessions could render positive results? Is this something for future research to explore?
· *This was a very heterogeneous group. Could future research benefit from further exploration by cultural/linguistic group? It is very possible that this works better in some groups than others – which could not be analyzed here.
Finally, feasibility studies are often accompanied by measures of acceptability. If interventions are not acceptable to the population, they will not be successful. The authors get at this a bit with attendance, but were any other standardized data collected on how participants felt about the intervention? If not, the last session had a “general feedback” component, no?
